# Kinetic Models of Discrete Opinion Dynamics on Directed Barabási–Albert Networks

**DOI:** 10.3390/e21100942

**Published:** 2019-09-26

**Authors:** F. Welington S. Lima, J. A. Plascak

**Affiliations:** 1Dietrich Stauffer Computational Physics Lab, Departamento de Física, Universidade Federal do Piauí, 64049-550 Teresina, PI, Brazil; 2Departamento de Física, Universidade Federal de Minas Gerais, C. P. 702, 30123-970 Belo Horizonte, MG, Brazil; 3Departamento de Física, Centro de Ciências Exatas e da Natureza, CCEN, Universidade Federal da Paraíba, Cidade Universitária, 58051-970 João Pessoa, PB, Brazil; 4Department of Physics and Astronomy, University of Georgia, Athens, GA 30602, USA

**Keywords:** non-equilibrium, phase transition, Monte Carlo simulations, Barabási–Albert networks

## Abstract

Kinetic models of discrete opinion dynamics are studied on *directed* Barabási–Albert networks by using extensive Monte Carlo simulations. A continuous phase transition has been found in this system. The critical values of the noise parameter are obtained for several values of the connectivity of these directed networks. In addition, the ratio of the critical exponents of the order parameter and the corresponding susceptibility to the correlation length have also been computed. It is noticed that the kinetic model and the majority-vote model on these *directed* Barabási–Albert networks are in the same universality class.

## 1. Introduction

There has been, recently, a great interest in the study of networks [1,2,3], which are different from the regular crystalline Bravais lattices, and are also frequently called scale-free networks. The treatment of such complex lattices was originally motivated by social organizations and computer connections, ranging from networks in nature to networks of people as well. However, spin systems defined on such complex networks have also been considered in the purpose to determine the character of its phase transition, if present, and the corresponding universality class in the case of critical behavior (for a recent review see reference [4]).

It should be mentioned that not only spin models have been considered on such networks. In the particular case of the *directed* Barabási–Albert networks (DBAN), it has been shown that the nearest-neighbor spin-1/2 Ising model has no phase transition [4,5,6]. Nevertheless, on these same networks the non-equilibrium majority-vote model (MVM) [7] presents a well-defined order–disorder dynamical phase transition [8].

Regarding models on networks, in 2012 Biswas, Chatterjee, and Sen [9] introduced a very interesting kinetic model of continuous (or discrete) opinion dynamics. The Biswas-Chatterjee-Sen (BCS) model has mutual interactions that can be both positive and negative, and a single parameter *p* that represents the fraction of negative interactions. Numerical simulations of the continuous version of the BCS model have indicated the existence of a universal continuous phase transition at a critical value $p=p_{c}$ with mean field exponents ν=2.00(1) for the correlation length, β=0.50(1) for the order parameter and γ=1.00(1) for the order parameter fluctuation (susceptibility).

The original BCS model has been defined on a fully connected graph of infinite range and, as said above, it can be treated in its continuum or discrete opinion dynamics versions. However, Mukherjee and Chatterjee [10] extended the model to square and cubic lattices, and the numerical results have indicated that the critical behavior of the BCS model on these lattices is the same as that of the Ising model in the same dimension. Thus, it should be interesting to study the BCS model on the complex DBAN and to compare the results to those obtained by the majority-vote model on the same network [8]. For a better comparison we will treat here the model within the discrete opinion dynamics version.

The plan of the paper is the following. In the next section, the BCS model in the discrete opinion dynamics, with the corresponding Monte Carlo simulations, and the thermodynamic quantities used to obtain the critical behavior are presented. The results are discussed in Section 3 and some conclusions are summarized in the last section.

## 2. Model and Simulations

### 2.1. Biswas-Chatterjee-Sen Model

The discrete version of the BCS model [9] on a DBAN can be described as follows. Consider a set of agents (individuals) which have, at time *t*, opinion variables $o_{i}\left(t\right)$, where *i* denotes the particular node of the DBAN with the network having a total of *N* agents. In the discrete version of the model, the opinion of an agent or individual *i* assumes only values ±1. In addition, the opinions change out of pairwise interactions via mutual influences/couplings $\mu_{ij}$ in such a way that we have
(1)$$\begin{array}{ccc} o_{i}\left(t+1\right) & = & o_{i}\left(t\right)+\mu_{ij}o_{j}\left(t\right)\;, \end{array}$$
(2)$$\begin{array}{ccc} o_{j}\left(t+1\right) & = & o_{j}\left(t\right)+\mu_{ji}o_{i}\left(t\right)\;, \end{array}$$
where *t* is the discrete time and the pairwise interactions between sites *i* and *j*, $\mu_{ij}$, and between sites *j* and *i*, $\mu_{ji}$, are quenched and integer variables as well, since they are associated with the links of the network, so they do not change in time. Due to the directed character of network links, the interactions $\mu_{ij}$ and $\mu_{ji}$ are not symmetrical and, in the most general case, can also assume zero value, stemming from the network construction.

It can be seen, in the above dynamics, given by Equations (Equation 1) and (2), that an agent *i* updates his opinion by interacting with agent *j*, and it is influenced by the mutual coupling term $\mu_{ij}$. As in the Ising model and MVM this discrete version of the BCS model presents the up-down symmetry with $o_{i}\left(t\right)=\pm 1$. In the present study, $\mu_{ij}$ is a discrete variable which takes the value −1 or +1 with probability *p* or 1−p, respectively. In other words, the disorder parameter 1−p denotes the fraction of positive pairwise interactions.

The ordering in this system can be determined by the average opinion over all individuals in the network, defined by
(3)$$O=|\sum_{i}^{N}o_{i}|/N.$$
By changing the fraction *p* of negative interactions in the fully connected graph of infinite range, one can observe, for $p<p_{c}$, a symmetry breaking ordered phase with a non-zero value of the order parameter *O*. On the other hand, for $p>p_{c}$, a disordered phase exists with O=0. At $p=p_{c}$ a continuous transition between the ordered and disordered phases takes place.

To study the critical behavior of the present model in the DBAN we can consider the same Ising-like quantities for the second-order phase transition. However, instead of a thermal driven criticality, here we have a configurational disorder driven transition. Thus, for the present model we are interested in the order parameter as a function of the disorder *p*, O(p), the order parameter fluctuations χ(p), which is the analogous of the magnetic susceptibility in spin models, and the corresponding reduced fourth-order Binder cumulant $U_{4}\left(p\right)$, where the last two quantities can be defined as
(4)$$\chi \left(p\right)=N\left(\langle O^{2}\rangle -{\langle O\rangle }^{2}\right),$$
(5)$$U_{4}\left(p\right)=1-\frac{\langle O^{4}\rangle }{3{\langle O^{2}\rangle }^{2}},$$
where 〈⋯〉 stands for configurational averages (and sample averages in our simulations, as is discussed below) computed at the steady states.

The above-mentioned quantities are functions of the disorder parameter *p* and, close to the transition point $p=p_{c}$, they obey the additional finite-size scaling relations
(6a)$$O\left(p\right)=N^{-\beta /\nu }f_{O}\left(x\right),$$
(6b)$$\chi \left(p\right)=N^{\gamma /\nu }f_{\chi }\left(x\right),$$
where ν, β, and γ are the usual critical exponents of the correlation length, order parameter and susceptibility, respectively, and $f_{O}\left(x\right)$ and $f_{\chi }\left(x\right)$ are the finite-size scaling functions with
(6c)$$x=\left(p-p_{c}\right)N^{1/\nu }$$
being the scaling variable.

The fourth-order Binder cumulant $U_{4}$ should be independent of (large) system sizes for |x|<<1, which allows one to estimate the critical value $p_{c}$. Therefore, from the size dependence of *O* and χ at $p_{c}$ we can obtain the exponents ratio β/ν (*O*) and γ/ν (χ).

It is also possible to evaluate the effective dimensionality, $D_{\mathrm{eff}}$, from the hyperscaling hypothesis
(7)$$2\beta /\nu +\gamma /\nu =D_{\mathrm{eff}}.$$

### 2.2. Simulations

We give below some details of the computer simulations. First, we initialize a directed Barabási– Albert network with only *z* neighbors which are connected with each other. Then, additional nodes are connected to this core following the Barabási–Albert prescription. In the end, we will reach a complete graph with a maximum value of N+z nodes. The links are directed by building a “Kertesz list”, where the node *j* can be in the neighbor list of node *i*, whereas the node *i* does not necessarily belong to the neighbor list of node *j*. After, the Monte Carlo simulations have been performed on DBAN with various values of the connectivity *z*, namely ranging from z=2 to z=100. In the present case, the Monte Carlo method is used in the evolution of the network together with the acceptance of the agents update. However, differently from the MVM that uses a master equation for its time evolution, here one uses the acceptance rules of the BCS model evolving on a Markov chain, which is a usual tool of the Monte Carlo approach.

For each connectivity *z* we have used system sizes with N=250, 500, 1000, 2000, 4000, 8000, and 16000 nodes. Although the relaxation time, and the number of Monte Carlo steps (MCS), as well as the number of independent runs to generate the statistics depend on the system size, we have taken, in general, for the larger networks, $1.5\times 10^{5}$ (MCS) for the system to reach its steady state, and then the time averages have been calculated over the next $3\times 10^{5}$ MCS. One MCS here is accomplished after *N* attempts to update the opinions of agents *i* and *j*, considering the evolution Equations (Equation 1) and (2). For all sets of parameters *p* and *z*, we have generated 100 distinct networks and, for each distinct network, we have simulated 100 independent runs. These quantities have shown to be enough to get data with reasonable error bars (as will become clearer below) and within our available computational time.

## 3. Results and Discussion

Figure 1 displays the fourth-order Binder cumulant $U_{4}$ as a function of the disorder parameter *p* for several DBAN with different number of nodes. In (a) the connectivity number is z=2 and in (b) we have z=100. It is clear from these figures that the system undergoes a second-order phase transition, since the cumulants tend to cross at the same value at the critical disorder parameter $p_{c}$ [11]. We can also note that the smaller the connectivity is, the more apparent the crossings are. Despite still having finite-size effect in the cumulants, the estimate of $p_{c}$ has been made by taking only the cumulant crossings of the larger nodes lattices (pairs of networks ranging from 2000 to 16,000 nodes). In the examples of Figure 1 we have $p_{c}=0.439\left(3\right)$ for z=2 and $p_{c}=0.388\left(4\right)$ for z=100. In the above data the errors are statistical ones. These, and other values of the critical disorder parameter for different connectivities, are given in the first row of Table 1.

In Figure 2 we have the ln-ln plot of the average opinion at the critical disorder, $O(p_{c})$, as a function of the number of nodes *N*, for several values of *z*. In this case, it is easy to see that, from Equation ([Disp-formula FD6a-entropy-21-00942]), the slope of the linear fit gives the critical exponent ratio β/ν. The corresponding results are reproduced in Table 1 together with the results for additional values of the connectivity *z*. From Figure 2 we can also notice that the critical exponent ratio β/ν changes as the connectivity *z* changes, implying in a sort of non-universal behavior. As we will discuss below, this is the general trend for the susceptibility exponent ratio as well.

Figure 3 displays the ln-ln plot of the order parameter susceptibility at the critical disorder value, $\chi (p_{c})$, as a function of the number of nodes *N*, for several values of the connectivity *z*. For the susceptibility, from Equation ([Disp-formula FD6b-entropy-21-00942]), the slope of the linear fit gives the critical exponent ratio γ/ν. One can clearly see that the exponent ratio γ/ν, contrary to what happened to the β/ν exponent, now increases as the number of nodes increases, implying, nevertheless, in a non-universal behavior for different values of *z*, as before. In this case, a stronger finite-size dependence of the critical exponent ratio can also be noticed for higher values of the number of nodes. However, instead of taking into account corrections-to-finite-size scaling, which further needs the knowledge of the (unknown) correction-to-scaling exponent, we opted to analyze the behavior of the fitted exponent by systematically dropping data from the smaller lattices until the large node sizes regime has been achieved. That is why, for *z* greater than 50, only the larger number of nodes (N≥1000) have been considered in the linear fit. The corresponding exponents are also shown in the first row of Table 1, including additional values of different parameters *z*.

The susceptibility, given by Equation ([Disp-formula FD6b-entropy-21-00942]), can still be used, at least in principle, to obtain the value of the ratio γ/ν by using its maximum value $\chi_{max}$ as a function of *p*, in this case at $p_{max}$. It turns out, however, that $p_{max}$ is of the order $p_{max}\leq 0.2$ for the connectivity in the range 2≤z≤100. This is a quite smaller value when compared to the $p_{c}\leq 0.44$ obtained above from the cumulant crossings $U_{4}$. In addition, even if we consider the maximum of the susceptibility in Equation ([Disp-formula FD6b-entropy-21-00942]) regardless the value of $p_{max}$, a different ratio exponent is obtained, as is depicted in Figure 4 for several values of *z*. One can see that γ/ν=0.128(5) obtained from the data with $p_{c}$ is much smaller than γ/ν=0.890(10) computed from the maximum value of the susceptibility.

One can clearly notice that not only $p_{c}$, coming from the maximum of the susceptibility, but also the corresponding critical exponent ratio γ/ν, are completely different from those values obtained from the fourth-order cumulant. It should be stressed that this very behavior is qualitatively the same as that obtained for the majority-vote model on the same DBAN [8]. However, the MVM on *undirected* Barabási–Albert networks (UBAN) exhibits a critical value of the connectivity and a corresponding critical exponent ratio which are of the same order whether estimated from $\chi_{max}$ or $\chi (p_{c})$ [8,12]. Although we do not have results of the BCS model on UBAN, the above results may be an indication that, independent of the dynamic model, the DBAN itself promotes this rather strange behavior of the order parameter fluctuation. This is further seen for other values of the connectivity *z*, as shown in Figure 5, and given in the first row of Table 1.

It is apparent from the results of Table 1 that both models have similar exponents ratio. In fact, in Figure 6 it is shown the critical exponent ratios β/ν and γ/ν, together with the effective dimensionality $D_{eff}/2$, as function of the connectivity *z*, with the data taken from Table 1 for both BCS model and MVM on the DBAN. It is clear from this figure that while the exponents ratio changes as *z* changes, the effective dimensionality remains the same and close to $D_{eff}\approx 1$. Although the critical exponents for both models are close to each other, as expected by the Grinstein criterion [13], the agreement is better for the effective dimensionality $D_{eff}$.

Figure 7 shows the phase diagram in the connectivity *z* and disorder *p* parameters plane for the BCS model compared to the MVM on the DBAN. In that scale we can notice that the non-universal critical disorder is indeed different for both models.

## 4. Conclusions

We have studied a discrete version of the non-equilibrium kinetic Biswas-Chatterjee-Sen (BCS) model through extensive Monte Carlo simulations on directed Barábasi-Albert networks (DBAN). In contrast to the Ising model, which has no phase transition on DBAN, the BCS model shows indeed a continuous phase transition. Our Monte Carlo simulations indicate that the critical exponent ratios β/ν and γ/ν change as the connectivity of the network changes, and are in the same universality class as the majority-vote model (MVM). However, the effective dimensionality $D_{\mathrm{eff}}$ is equal to unit, for all values of *z*, implying that the hyperscaling relation 2β/ν+γ/ν=1 may be valid. A similar behavior has been found in references [8,12,14] for the MVM, even regarding the non-expected behavior of the maximum value of the order parameter susceptibility and its corresponding critical exponent ratio. This fact also indicates that the DBAN itself may be responsible for this strange behavior regarding the susceptibility quantity on this dynamical model. Of course, the study of other dynamic models defined on the DBAN would be very welcome to check this important point. 

## Figures and Tables

**Figure 1 entropy-21-00942-f001:**
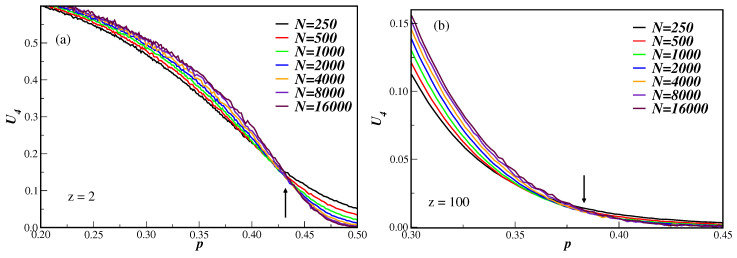
Fourth-order Binder cumulant $U_{4}$ as a function of the disorder parameter *p* for several number of nodes *N* and two connectivity numbers: z=2 (**a**) and z=100 (**b**). The vertical arrows indicate the corresponding estimate of $p_{c}$, which is given in Table 1. For clarity, only the general trend of the cumulant behavior is shown without the error bars.

**Figure 2 entropy-21-00942-f002:**
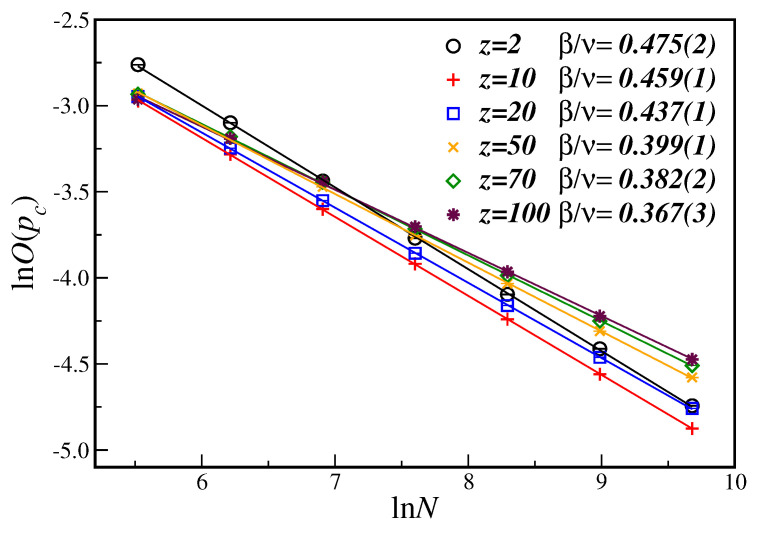
(Color on-line). Ln-ln plot of the average opinion at the estimated critical disorder $O(p_{c})$ as a function of the number of nodes *N* for different connectivities *z*. The lines are the best linear fit with the slope being the critical exponent ratio β/ν. Please note that the shown error bars are smaller than the symbol sizes.

**Figure 3 entropy-21-00942-f003:**
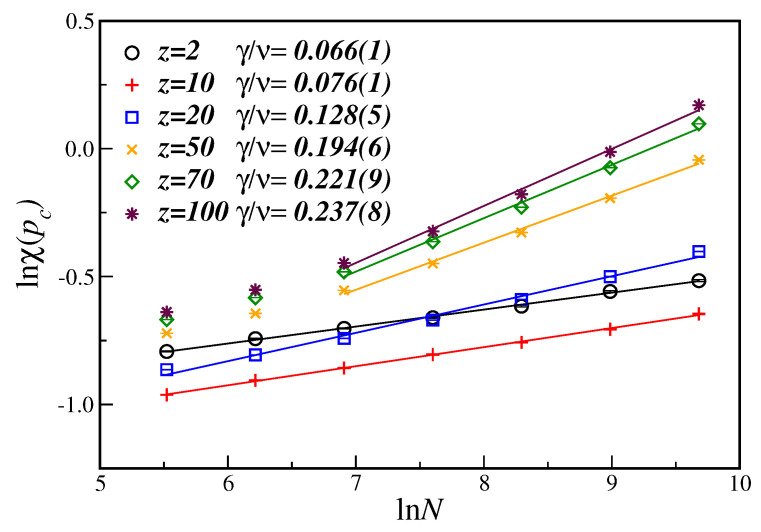
Ln-ln plot of the susceptibility $\chi (p_{c})$ at the estimated $p_{c}$ as a function of the number of nodes *N* for different connectivities *z*. The lines are the best linear fit with the slope being the critical exponent ratio γ/ν. The displayed error bars are smaller than the symbol sizes.

**Figure 4 entropy-21-00942-f004:**
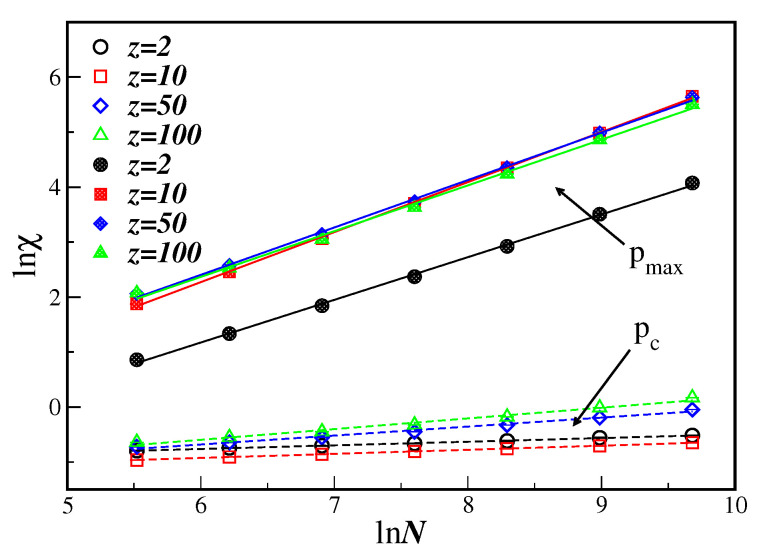
Ln-ln plot of the susceptibility χ as a function of *N* for several values of the connectivity *z*. Open symbols correspond to the susceptibility evaluated at $p_{c}$, while full symbols are evaluated at the maximum value of the susceptibility $p_{max}$. The lines are linear fits with the corresponding exponents ratio given in Table 1.

**Figure 5 entropy-21-00942-f005:**
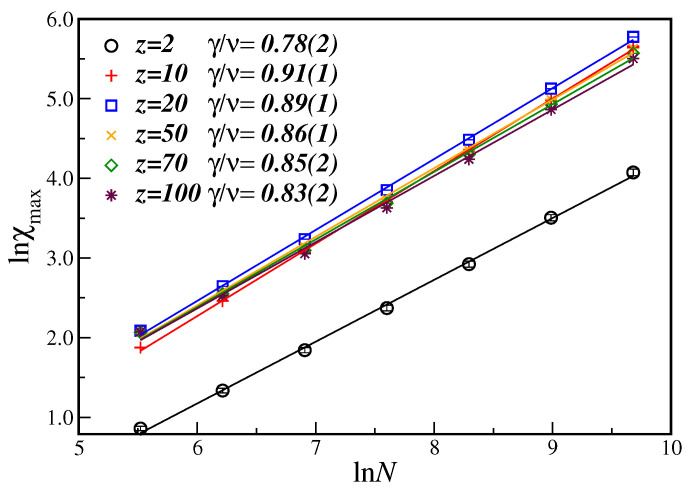
Ln-ln plot of the maximum of the susceptibility $\chi_{max}$ at $p_{max}$ as a function of the number of nodes *N* for different connectivities *z*. The lines are the best linear fit with the slope giving the critical exponent ratio γ/ν.

**Figure 6 entropy-21-00942-f006:**
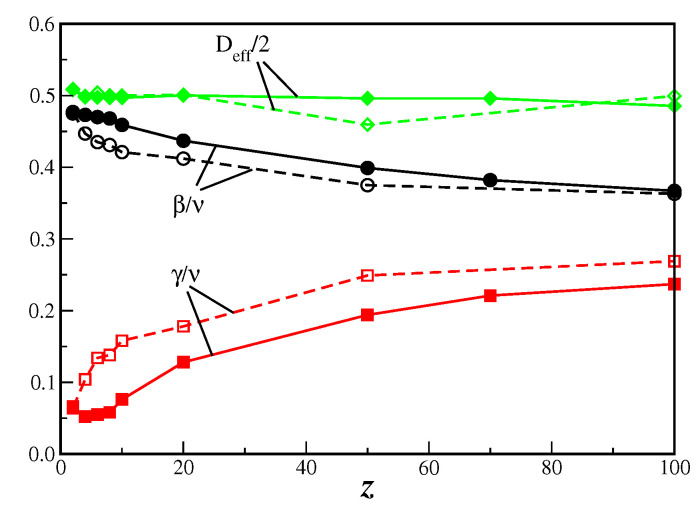
Critical exponents ratio β/ν and γ/ν, and half value of the effective dimension $D_{eff}$ as a function of the connectivity *z*. Full symbols correspond to the present BCS model, and open symbols to the MVM [8], both on the same DBAN. Full and dashed lines are only guide to the eyes.

**Figure 7 entropy-21-00942-f007:**
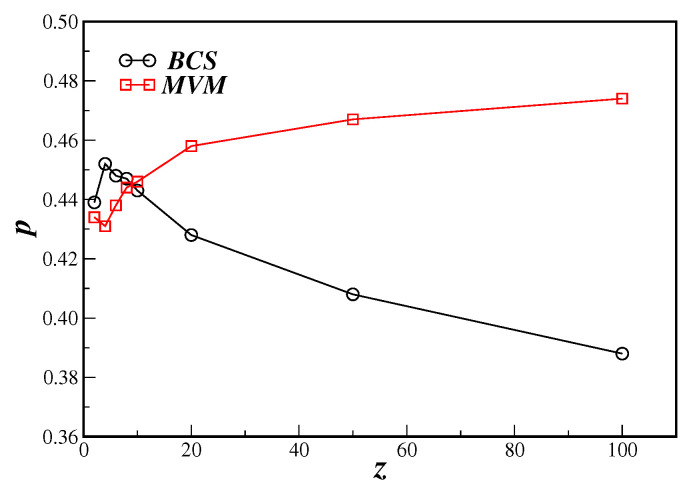
Phase diagram in the connectivity *z* and disorder *p* parameters plane for the BCS model (circles) and MVM (squares) on the DBAN.

**Table 1 entropy-21-00942-t001:** The critical connectivity parameter $p_{c}$, the critical exponents ratio β/ν and γ/ν, and the effective dimension $D_{\mathrm{eff}}$ on *directed* BA networks for different connectivities *z*. For each entry of the connectivity *z*, the upper row corresponds to the results for the discrete BCS model, while the lower row corresponds to the results for the MVM. The ratio γ/ν is obtained from two estimates: at $p_{c}$ and at $p_{max}$. The former one comes from the values of χ at the critical concentration and the latter one from the maximum value of χ as a function of *p*. Error bars are statistical only.

*z*	$p_{c}$	β/ν	$\gamma /\nu \;(p_{c})$	$\gamma /\nu \;(p_{\max})$	$D_{\mathrm{eff}}$
2	0.439(3)	0.475(2)	0.066(1)	0.775(12)	1.016(4)
	0.434(3)	0.477(2)	0.064(12)	0.895(10)	1.018(9)
4	0.452(3)	0.473(1)	0.052(1)	0.829(10)	0.995(5)
	0.431(3)	0.447(1)	0.104(2)	0.888(9)	0.998(3)
6	0.448(3)	0.470(1)	0.055(1)	0.845(9)	0.995(6)
	0.438(2)	0.435(2)	0.134(5)	0.861(3)	1.008(6)
8	0.447(3)	0.468(1)	0.058(1)	0.860(9)	0.994(5)
	0.444(5)	0.431(1)	0.138(2)	0.851(5)	1.000(2)
10	0.443(3)	0.459(1)	0.076(1)	0.909(8)	0.994(3)
	0.446(3)	0.421(2)	0.158(3)	0.834(7)	1.000(5)
20	0.428(3)	0.437(1)	0.128(5)	0.890(10)	1.002(3)
	0.458(4)	0.412(1)	0.178(2)	0.795(11)	1.002(2)
50	0.408(2)	0.399(1)	0.194(6)	0.861(14)	0.992(9)
	0.467(2)	0.375(4)	0.249(7)	0.735(17)	0.999(11)
100	0.388(4)	0.367(3)	0.237(8)	0.832(20)	0.971(31)
	0.474(3)	0.363(4)	0.269(5)	0.674(23)	0.999(9)

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
