# Peer review of "Kinetic Models of Discrete Opinion Dynamics on Directed Barabási–Albert Networks"

_entropy, 2019, doi:10.3390/e21100942_

Round 1

Reviewer 1 Report

Authors present results from Monte Carlo simulations on the directed Barabasi-Albert network. The modelling works includes a systematic study over the network parameters, especially size and connectivity, that affect the order parameter quantified through the critical exponents.

The work is interesting, it has the potential to draw general readership and the topic shows increased popularity. However, revisions are necessary for the manuscript to be in publishable form in Entropy. Below is my list of comments and questions.

) One of the main findings is the discrepancy observed between the critical exponent as predicted from p_c and from p_max. I think the result unexpected and confusing. Deeper analysis and interpretation is needed on this trend  which appears to be systematic over the whole range of variables studied.

) In Fig. 3 the behaviour of small-N seems to deviate from the linear trend established for large-N where the slope has been calculated. The authors should comment on this and if possible provide an explanation.

) Curves of Figure 1 are obtained as averages over the 100 distinct runs and the authors avoid to show the error bars for clarity, am I correct? “ As a question of clarity” could be changed to “For clarity”.

) Figures 2, 3 and 5 show various curves for different connectivities starting from z = 4. Given that results are available also for z = 2 it would be preferable to show the results corresponding to the lowest limit (z = 2) instead of z = 4.  

) Figure 4. Perhaps the corresponding points for another connectivity (apart from z = 20) could be added for both sets.

) Line 115: “is a rather larger”. Do the authors mean “smaller” ?

) In Table 1 p_c and p_max (labels in columns 4 and 5) could be included in parenthesis as in the present form they seem part of the formula.

) Some plots appear in log-log others in ln-ln. While the general trends are the same it is better to have consistency.

) Some grammar and syntax errors:

Line 76: “system reach” -> “system to reach”

Line 92: “first row of Table 1”. Do the authors mean “first column” ? Or they want to differentiate with the MVM results? If it is the latter it should be clarified in the sentence.

Line 119: “x(p_c” -> “x(p_c)”.

Line 132: “Fig. 6” -> “Fig. 7”.

Line 138: “exponents ratio .. and .. changes” -> “exponent ratios .. and .. change”.

Author Response

We are very grateful for your suggestions. Now we think our manuscript is much better than before.              

Reviewer 2 Report

The manuscript presents simulation study on a model of opinion dynamics proposed by Biswas et al. placed on directed Barabasi-Albert networks. The work focuses on the critical phenomena exhibited by the model. The authors determine the transition points based on two methods (by the use of Binder cumulant and the maximum value of fluctuations) and the critical exponent ratios. The results are compared with the majority-vote model, and it turns out that both models fall into the same universality class.

The manuscript is well organized and clearly written. The results are presented neatly, and the simulations are describe in detail. Moreover, the raised problem, that is, studying the impact of the network topology on universality classes, is of current interest. In fact, I really enjoyed reading the paper.  The study creates further research opportunities as well. For instance, it would be interesting to analyze the model on other scale-free networks and determine the influence of the scale-free network exponent on the critical behavior of the system since it is known that in the case of the Ising model it plays a great role.

I have only several minor comments:

In relation to the statement in the second paragraph: “This is a clear example that different models will in fact exhibit different transition behaviors when defined in the same complex lattice.” - why would anyone suppose that different models should exhibit the same behavior on the same networks. I would rather assume different behavior by default. For, instance, it is know that the q-voter model displays continuous phase transitions on undirected BA networks, whereas Ising model exhibits transitions of infinite order in this case. In section 2.1, we have two equations for the time evolution of opinions o_i and o_j – Eq.(1) and Eq.(2). In the second one, Eq.(2), I would swap the indexes in order to have \mu_ji instead of \mu_ij. In fact, we do not necessarily have to have symmetric interactions, so \mu_ji may not be equal to \mu_ij. I think I would make the description more general. Actually, since in this work the undirected networks are considered, \mu is not symmetric. Moreover, we can even say that one of the \mus, \mu_ij or \mu_ji is 0, which stems from the network construction. Maybe, I would also consider emphasizing that \mus are quenched, that is, they are associated with the links of the network, so they do not change in time. I believe it would make the description more transparent since the annealed version, in which \mus vary in time, is possible as well. Regarding the description of simulations, I suggest specifying the initial conditions for the networks. The growing mechanism – the preferential attachment – is clear, but how does the network look like at t=0? Since the preferential attachment is considered, the process cannot start with unconnected z nodes. Is it a complete graph with z nodes? but then how are the links directed? In conclusions (line 137), the term “continuous dynamic phase transition” is used. How is “dynamic” understood in this context? In line 16, the right parenthesis is missing at the end of the sentence. In line 84, we can read “DBAN networks”, but N in the acronym already stands for a network – so I guess it is a kind of repetition. In line 102, after Eq., the parentheses are doubled. In line 119, the right parenthesis is missing at the end of the sentence. At the beginning of line 130, there is a reference to Fig. 6 instead of Fig. 7.

Author Response

(The authors gave the same response as above.)

Round 2

Reviewer 1 Report

The revised manuscript will make a strong contribution to the Entropy journal. I have only some very minor comments that the authors could consider adopting before publication.

) Given that Monte Carlo method is employed for the evolution of the network it could be good if the authors could provide the acceptance criterion used to update agent opinion.

) Line 79: “node i is does not belong”. Is this or “node i does not necessarily belong” ?

) Line 55: “the discrete value -1” -> “the value -1”

) Fig. 1 legend: “which are given” -> “which is given”.

) Line 118: “regime have been” -> “regime has been”.

) Table 1: “the up row are” -> “the upper row corresponds” ; “the down row are” -> “the lower row corresponds”.

Author Response

Dear Reviewer,

Thanks, we are very grateful for your work as a proofreader of this manuscript.
